# 3D-Printed Graphene-Based Bow-Tie Microstrip Antenna Design and Analysis for Ultra-Wideband Applications

**DOI:** 10.3390/polym13213724

**Published:** 2021-10-28

**Authors:** Emine Avşar Aydın

**Affiliations:** Department of Aerospace Engineering, Adana Alparslan Türkeş Science and Technology University, Adana 01250, Turkey; eaydin@atu.edu.tr

**Keywords:** antenna performance, bow-tie antenna, bandwidth, graphene, chemical potential, 3D printer

## Abstract

In this study, the effects of graphene and design differences on bow-tie microstrip antenna performance and bandwidth improvement were investigated both with simulation and experiments. In addition, the conductivity of graphene can be dynamically tuned by changing its chemical potential. The numerical calculations of the proposed antennas at 2–10 GHz were carried out using the finite integration technique in the CST Microwave Studio program. Thus, three bow-tie microstrip antennas with different antenna parameters were designed. Unlike traditional production techniques, due to its cost-effectiveness and easy production, antennas were produced using 3D printing, and then measurements were conducted. A very good match was observed between the simulation and the measurement results. The performance of each antenna was analyzed, and then, the effects of antenna sizes and different chemical potentials on antenna performance were investigated and discussed. The results show that the bow-tie antenna with a slot, which is one of the new advantages of this study, provides a good match and that it has an ultra-bandwidth of 18 GHz in the frequency range of 2 to 20 GHz for ultra-wideband applications. The obtained return loss of −10 dB throughout the applied frequency shows that the designed antennas are useful. In addition, the proposed antennas have an average gain of 9 dBi. This study will be a guide for microstrip antennas based on the desired applications by changing the size of the slots and chemical potential in the conductive parts in the design.

## 1. Introduction

Microstrip patch antennas are used in aviation and satellite communication, biomedical, radio, and wireless applications due to their important features such as reduced size, weight, and cost [1]. However, low efficiency and narrow bandwidth also can give rise to significant disadvantages [2]. Among the simplest methods of overcoming these deficits, the best-known methods increase the substrate’s thickness and the conductive elements to be used [1,2,3]. The patch antennas typically have a narrow bandwidth, and using a thicker dielectric substrate widens it [4]; for achieving a bandwidth greater than 10%, a capacitive probe is generally needed [5,6]. Moreover, bandwidth optimization can be achieved with thick substrates or stacked patch configuration, but these also cause noise. Unfortunately, stacked patch solutions can be obtained with a complex fabrication process, and with these methods, the layers’ alignment can cause some problems, such as air gap formation [7,8,9].

The compactness of the microstrip antennas is an important feature [10]. Typically, patch antenna size is determined by its resonant size. However, it can be very large for some antennas in practical applications, especially at low frequencies, and it can increase the wireless device’s size. Short-circuit walls [11] and pins [12], folded patches [13], or substrates with high dielectric permeability [14] can be used to produce compact antennas. However, shorting pins and walls require complex realization and can generate non-symmetrical broad-edged radiation [15]. Folded patches can also be difficult to apply. Finally, materials with a high dielectric constant have insufficient radiation, and they reduce the bandwidth and are expensive.

Previously, two physicists at the University of Manchester in 2004 discovered that graphene could be obtained by isolating a one-atom-thick layer from graphite. The second one is a simple two-dimensional sheet shaped in a hexagonal pattern consisting of carbon atoms [16]. Since its discovery, graphene has been the material that has been studied by many researchers from different fields due to its many new and unique physical properties. Moreover, apart from having electrical mobility higher than 2.105 cm^2^Vm, it has a unique quantum Hall effect, good flexibility, and excellent mechanical resistance, which provides many advantages for different areas such as nanoelectronics, material science, photovoltaics, and engineering. Another important feature is that it has a thermal conductivity of more than ten times copper [17]. Additionally, graphene is already used in biotechnology, antibacterial materials, disease diagnosis, drug delivery, and cancer targeting [18,19,20].

A reconfigurable antenna is defined as an antenna in which at least one of its features can be changed by external intervention after its production is completed. There are many ways of classification of these reconfigurable antennas. Here are a few examples in order: It can be listed as causing the antennas to be changed again (changing the current lines, changing the dielectric properties of the antenna elements, geometric deformation), having the elements that allow the change (diodes, transistors, etc.) or changing the geometric structures of the antennas again. Various techniques are used in antenna reconstruction methods. Some researchers use localized active components that allow changing half-point current or impedance lines [21], others apply mechanical remodeling in the antenna [22], and still, other researchers apply configurations using substrates with tunable properties [23]. Again, some techniques go the way of reconfiguring power supply networks or opting for excitation antenna arrays [24]. Developments in microelectronics, the discovery of electrically operable switches, and varactors have revealed important suggestions for the emergence of reconfigurable devices. However, the elements to be put into the configuration increase the device size and bring additional production costs. During the reconstruction process, the electromagnetic properties of an antenna can be changed by applying the properties of the materials in the antenna structure (such as permeability, electric–magnetic field effects) with external control without adding any elements. These antenna structures are mostly used in the patch or substrate parts that make up the antennas. Graphene is a great material because it has excellent optical and electronic properties [25,26,27] and also because its resonance properties can be easily changed by applying an external voltage [28,29,30].

Using graphene provides a great advantage to antenna designs because it has an efficient dynamic tuning [31,32]. The frequency-dependent surface conductivity of graphene can be easily changed by causing changes in its chemical potential due to applying a gate voltage. While this important feature of graphene provides an advantage in adjusting the resonance frequencies of antennas, this situation provides the chance to easily manufacture antennas that can be used, especially at microwave frequencies.

The frequency-dependent two-dimensional surface conductivity of single-layer graphene is expressed by the Kubo formula (Equation (1)) [33]:(1)$$\sigma (\omega ,\mu_{c},\Gamma ,Τ)\approx -j\frac{q_{e}^{2}k_{B}Τ}{\pi h^{2}(\omega -2j\Gamma )}\times \left(\frac{\mu_{c}}{k_{B}Τ}+2\ln(e^{-\frac{\mu_{c}}{k_{B}Τ}}+1)\right)$$

ω: the angular frequency$\mu_{c}$: the chemical potentialΓ: the scattering rateΤ: the temperatureτ: the relaxation time$k_{B}$: the Boltzmann$h^{2}$: the reduced Planck constant.

The effect of changing graphene’s chemical potential ($\mu_{c}$) on the surface conductivity has been studied [34]. This effect varies depending on the carrier density, which can be controlled by gate voltage, electric bias field, or chemical doping. Increasing the chemical potential also causes an increase in graphene surface conductivity. In other words, this causes a shift in antenna resonances at higher frequencies. On the other hand, it allows the emergence of flexible antenna designs that can be changed by the shift in antenna resonance caused by the chemical potential change. The value in chemical potential can be controlled electrically by varying the gate voltage ($V_{g}$). The formula explaining this situation is given below (Equation (2)). Graphene conductivity ranges are given according to the relevant frequency bands in the study of Gatte et al. [34].
(2)$$V_{g}=\left[\frac{q_{e}\mu_{c}^{2}h}{\pi \hbar^{2}v_{f}^{2}\varepsilon_{0}\varepsilon_{r}}\right]$$

Inum et al. give the real and imaginary values of the surface conductivity of graphene depending on the frequency for a changing chemical potential and a constant relaxation time at their study [33]. They show that the chemical potential and the conductivity change point are important parameters to adjust the resonance frequency.

In a study [35], the antenna produced with graphene was compared with the other antennas with different metals instead of graphene. When the results are examined, it is seen that the reflection coefficients of the metal-based antenna are less adapted than the graphene. On the other hand, graphene performs better in terms of reflection coefficient, decreasing from −6.440 to −15.125 dB. Graphene shows better results than metal in terms of performance with a well-adapted resonance frequency. Prasanna and Banu [36] examined the “Effect of Copper and Graphene Material Bow-tie Structured Antenna for 1.2 GHz Application.” It is seen that graphene performs better in antennas, since its conductivity is much better than copper. In addition, resistance and stub-matching problems are significantly reduced compared to copper and offer better bandwidth. All these situations suggest that using graphene is more functionally effective than copper. Considering all these parameters, graphene was used as a conductive material to produce the antenna suitable for the desired target both at an affordable cost and in the desired bandwidth, and a 3D printer was used as the production technique.

A traditional microstrip patch antenna has a high-quality factor, leading to some disadvantages related to the narrow bandwidth. Many techniques have been developed to increase the impedance bandwidth, including adopting the known stacked patches [37], parasitic resonators [38], and capacitive coupling feed [39,40]. It is still challenging to obtain a wider bandwidth, since the main solution uses a thick and/or low dielectric constant substrate.

Metamaterial-based absorbers and antennas with different fractal structures are being studied to expand the bandwidth [41,42]. While studies to increase the bandwidth of antennas are important, it is also very important that antenna production costs are affordable and easy to manufacture.

It is known that all conventional antenna manufacturing techniques have disadvantages such as high tool and equipment cost, small volume prototyping, or long lead times for production [43,44,45,46]. The new technologies used for antenna prototyping are laser micromachining and 3D printing technologies. The laser micro-processing technology is a common method that can be used to process many materials such as plastic, glass, and metal and has a versatile process to produce the desired outputs by providing thin foils. This technology includes many mechanisms such as cutting, drilling, marking, turning, and threading. The ability to drill holes at the micron level or make fine cuts is preferred in antenna production as in many areas. Apart from that, this technology presents some problems as it has a thermally induced process. For example, it causes a change in material properties in the heat-affected area, as well as causing thermal stress. In addition, they are economically very expensive and have optical problems. Therefore, laser micromachining technology was not preferred in this study.

Recently, the expansion of 3D printing offers new chances for low-cost, fast, and on-demand production of millimeter-wave, microwave, and terahertz (THz) antennas and components [47,48,49,50]. With the rapid development of 3D-printing technology, an alternative solution has been provided, especially for the application of microwave antennas [51,52]. It offers higher manufacturing accuracy at a lower cost and provides great flexibility in design and production. Therefore, it has many advantages compared to the standard production methods. With 3D printing, the devices can be produced in one piece, and the performance loss caused by combining the parts can be prevented.

There are two widely used 3D-printing techniques [53,54,55,56,57,58,59,60,61]: polymer/dielectric and all-metal. As the working frequency band enters the millimeter/microwave range, the printed parts’ quality for the resulting antenna is of great importance. The print resolution, structural limitations, and surface quality are among the many intangible factors that are hugely important for a functional antenna part. Millimeter-wave/microwaves/THz antennas usually have great details in their geometry, which poses challenging fabrication methods. For all these reasons, it has become attractive to use commercial 3D printers to achieve high spatial resolution, small structure layer thickness, and low surface roughness.

Microstrip antennas have become remarkably interesting in that they can be used in many different application areas owing to their wide operating bandwidth and easy adjustment alternatives [62,63,64,65,66]. Due to advances in handheld electronic devices, these devices need to be thinner to improve performance in small-sized devices [67,68,69]. Until now, in the literature, many antenna studies with different designs—the circle [70], ellipse [71], triangle [72], fractal [73], U-shaped [74], etc. [75,76,77]—have been carried out to increase the bandwidth and miniaturization. The bow-tie antenna, one of the microstrip antenna types, has also been studied in many different applications.

It is well known that the bow-tie antenna designs are application-specific, and the slots are added to increase the bandwidth. Therefore, two parameters need to be well specified for better performance: the location and size of the slot. One of the contributions of this study is that the slot dimensions are chosen especially for the required bandwidth and gain.

Graphene-based antenna studies have been carried out intensively, especially for the last five years [33,35,78,79,80,81,82,83,84,85,86]. However, using graphene as an antenna is not always an easy task, and the production techniques also play an essential role in this manner. Another contribution of this study is that a standard and cheap 3D printer is used to produce the antennas. In addition, the study results can be used as a practical reference of the effect of the slot size on the antenna performance and can be considered the novelty of this study. Finally, the graphene-based patch antenna’s gain is better than the copper antennas when used as the radiating element.

## 2. Material and Method

### 2.1. Materials

In this study, Esun PLA Plus Filament (Filament, esun, Shenzhen, China) was preferred for the substrate material of the antenna due to its high surface quality, high layer structure, and ten times more durability than standard PLA filaments. Detailed information on Physical and Print performances is given in Table 1.

Black Magic 3D Conductive Graphene Composite filament is used for the conductive part of the antenna due to its superior conductivity and improved mechanical properties, and its detailed properties are given in Table 2.

The bow-tie microstrip patch antenna design procedure is given in Figure 1. Computer analysis software is central to this work for the design process, as the researcher must have the appropriate design parameters to start the production process. Simulation software is an essential tool for designing a microstrip patch antenna with bandwidth at the desired resonance frequency and showing band-pass, band-stop, or ultra-wideband characteristics. Otherwise, a new antenna must be produced for every changed parameter of the antenna, and the reflection/transmission be measured. Therefore, this process causes a loss of time and increases the design cost. For this reason, using simulation and design software to obtain the transmission/reflection characteristics and optimize the antenna parameters reduces the cost and saves time. In this study, software named CST Microwave Studio was used for microstrip patch antenna design. Polylactic Acid from Esun (Esun PLA+) with a relative dielectric constant of 2.54 and a loss tangent of 0.015 was chosen as the suitable substrate material for the designed antenna. The dielectric material thickness was determined as 1.70 mm. Graphene PLA from BlackMagic was preferred for the conductive part, and its thickness is also 1.70 mm. A computer code that performs the mathematical calculations of the bow-tie antenna was written using MATLAB software (2019, The Mathworks, Inc., Natick, MA, USA). According to the calculated dimensions, the design of the antenna was carried out with the CST Microwave Studio software (CST Studio Suite 2019, Dassault Systèmes, Waltham, MA, USA).

In this section, the parameters affecting the configurability of the graphene-based antenna are evaluated. The first of the two kinds of parameters are related to the geometric structure of the antenna, and it can be optimized with numerical simulations, while the other is related to the graphene model related to the calculation in the previous section. The study aims to understand the behavior of the graphene-based antenna and contribute to future studies of graphene-based antennas.

### 2.2. Design and Production of the Bow-Tie Microstrip Antenna

A bow-tie microstrip antenna structure is planned to improve the antenna’s performance in biomedical applications. The design and simulation study of the proposed antenna is carried out in the CST Microwave Studio. The antennas are designed in Tinkercad, which is an online 3D design platform developed by Autodesk Inc. (San Rafael, CA, USA) and built with Creality Ender 3 Pro 3D printer (Creality, Shenzhen, China).

Material selection is one of the pivotal problems to be considered in the design process, as it has a significant impact on antenna performance. Polylactic Acid (PLA; ε_r_ = 2.54 and tanδ = 0.015) and Acrylonitrile Butadiene Styrene (ABS) are effective on the antennas’ performance. They are frequently preferred in the production of dielectric parts of antennas and microwave components. On the other hand, graphene is a suitable material for producing the conductive part of Microwave and Radio Frequency (RF) products due to its very good properties such as high charge mobility, zero bandgap, high heat conduction, high surface area, and excellent biocompatibility. Composite Conductive Graphene PLA filaments have recently been introduced, allowing graphene to be used with a 3D printer [84]. In this study, the designed antennas are produced using Conductive Graphene PLA filament for the core of the antenna and standard PLA for the dielectric substrate parts. The volume resistivity of the used graphene filament is 0.6 Ω-cm [89]. The product of the sheet resistance (Ω/sq) and the thickness of the material in centimeters is equal to the volume resistance (Ω-cm) [90,91,92], and therefore, the sheet resistance can be obtained as given in Equation (3).
(3)$$\begin{array}{l} R=\mathrm{the}\;\mathrm{volume}\;\mathrm{resistivity}/\mathrm{thickness}\;\mathrm{of}\;\mathrm{graphene}\;\mathrm{sheet}\\ R=0.6\;(\Omega -\mathrm{cm})/0.17\;(\mathrm{cm})\;=\;0.359\;\Omega \end{array}$$

The radiating patch of the microstrip antenna may be square, rectangular, circular, elliptical, triangular, or any other shape [93]. Generally, triangular structures or curved structures provide better impedance matching. The size and shape of the patch are some of the most determining factors in the properties of the antenna. However, the change of shape does not bring significant changes in the radiation characteristics. The half-power beamwidth is generally 70–90 degrees, and antenna gain is 5–6 dB. By choosing the patch model and shape appropriately, a design can be made in accordance with the desired resonance frequency, pattern, impedance, and polarization. Another novelty of this work is that the slot sizes have been chosen specifically for the required bandwidth and gain. Slots are used to regulate the current path and shape the electric field. For slots, sharp points allow electric field lines to break more easily. Circular slots show an improvement in antenna performance, which is weaker than square and rectangular slots. Filtering is done for a frequency when a slot is opened in a broadband patch made for ultra-broadband. Therefore, the optimum values of the antennas, whose numerical analyses were performed by using the finite integration technique in the CST Microwave Studio simulation program, were used in the designs. In addition, the effect of slots and their different sizes on antenna performance is shown.

In this study, three different antennas, one without a slot, Figure 2, the other two with different patch sizes, Figure 3, are designed and produced. The dimensions of the antennas are given in Table 3.

The three bow-tie antennas produced for use in biomedical applications are shown in Figure 4. If it is desired to be used in more specific applications, it can be easily adjusted with antenna design changes.

An SMA connector is connected to the antenna’s terminals by heating and softening the graphene; see Figure 4. The connector’s positive pin is soldered to the right triangular patch that acts as a positive terminal, and the negative pin is soldered to the left triangular patch.

The microwave antennas’ design and dimensions were optimized through the simulations using the CST Microwave Studio software. The CAD design was prepared using Tinkercad, and Ultimaker Cura was used to generate gcode for the 3D printer. A commercial Creality Ender 3 Pro printer was utilized to manufacture a 3D printed substrate part from PLA and a conductor part from graphene. The bed and the nozzle temperatures were adjusted as 60 °C and 205 °C, respectively for the PLA and 50 °C and 220 °C for the graphene. An intermediate printing head speed of 50 mm/s and 30 mm/s was selected for PLA and graphene, respectively. A nozzle with 0.5 mm size is used for the printing. Figure 5 shows the manufacturing of the antenna with a commercial 3D printer. Antenna measurements were made in the operating frequency range of 2–20 GHz using a PNA-L Agilent vector network analyzer. Before the measurement was carried out, the VNA was calibrated with a calibration kit with a short circuit, open circuit, and loading apparatus, respectively. After the calibration process, the produced antennas were connected to the VNA, and the return loss parameter and gain measurement results were obtained.

In addition, it was also investigated how the changes to be made in antenna geometries, especially in the graphene patch part, can affect the cross-sectional absorption areas. The physical parameters for all antennas are assumed to be T = 300 K and $\mu_{e}$ = 0 eV. It is seen that the antennas have resonances close to each other, and there is little difference between them, resulting from different patch structures. As shown in Figure 6, it is clear that the antenna has a large $\sigma_{abs}$ value, especially in the patch part, which means there is no inference.

## 3. Results and Discussion

Before interpreting the results, the measurement device and its arrangement mentioned in the previous section are shown in Figure 7. The transient solver of the simulation program was used to obtain an accurate result, and the necessary number of network applications was provided. The return loss of the antenna was measured with the aid of a network analyzer.

The reflection coefficients of Antenna-1, Antenna-2, and Antenna-3 are given in Figure 8a–c, respectively, to show both the advantage of the adjustable surface conductivity of graphene and to examine its usefulness in microwave antenna designs. Analyses were carried out by applying electrical fields of 0.1, 0.3, 0.5, and 0.7 eV, respectively, to the antenna created in the CST Microwave Studio software environment. These data presented in Figure 8 prove the advantages of highly variable graphene conductivity for tuning the antenna resonance frequency, which is also mentioned in the graphene material part. Accordingly, the relaxation time (τ) is accepted as 0.1 ps. When the data in the figure are examined, there is an observed improvement in harmony of the chemical potential for Antenna-1 and the reflection coefficient, while inconsistency is observed for Antenna-2 and Antenna-3, especially at 0.7 eV.

It has been shown that the polarization and permeability of the antennas can be achieved without making any changes in the antenna dimensions or physical structure only by changing the voltage applied to the graphene layer, which changed the Fermi energy level [94,95,96,97]. Equation (1) shows that the conductivity of graphene can be adjusted dynamically by changing its chemical potential. Figure 9a–c represent the actual and simulated values of graphene surface permeability for Antenna-1, Antenna-2, and Antenna-3, respectively, when the chemical potential is changed between 0 and 1.0 eV.

Figure 10 shows the bandwidth performance of the antennas in terms of return loss along with the lower ($f_{L}$) and upper ($f_{H}$) frequencies, where the return loss curve exceeds −10 dB. While it is seen that Antenna-1 provides the best performance in terms of return loss bandwidth, other antennas perform quite well in wideband ranges. In Figure 11, normalized radiation patterns were measured in both the E-plane and the H-plane, and the results were quite favorable.

Although the results of the simulation and experimental studies are generally compatible, minor inconsistencies may occur. Simulation environments are ideal environments, and the values of materials used in simulation environments are average values. There are slight differences between the values of PLA and graphene material used in the simulation environment and the materials used. In addition, minor faults caused by production, such as scratches or difficulties in soldering the SMA connector, can lead to different results. Another important situation is that this study used a graphene filament for antennas instead of the raw graphene material.

The comparison for efficiency and performance of graphene-based bow-tie antennas with previous ultra-wideband antennas are given in Table 4. This table shows that the graphene-based bow-tie antennas designed in this study perform quite well compared to other studies in the literature. The results obtained from these three proposed antennas show that they can operate in a wide bandwidth of 2–20 GHz. It has been proven that the geometric changes made in the conductive parts of the antenna also affect their performance.

In order to demonstrate the effect of graphene in one of the graphene-based antenna studies, the study was carried out by choosing copper as the conductive patch part [101]. From the study of Azizi et al., it is clear that the return loss peak value is almost twice that obtained with the copper patch (about −29 dB) for the graphene patch [101]. In another study [34], undoped pure graphene is used for the antenna design, and the antenna is modeled in the simulation. Then, different graphene models were created, and the corresponding surface conductors were obtained by applying voltage or adding chemical additives. To test and compare the graphene antenna performance, also, a traditional copper antenna design is prepared. The results showed that the graphene antenna has significant improvements in good reflection, radiation efficiency, and gain compared to the copper antenna. Furthermore, simulation studies for the Antenna-1 structure in this study were repeated for the case in which the conductive part is graphene and copper. As can be seen in Figure 12. Although the geometry is the same, it is seen that the selection in the conductive part affects the antenna performance.

Additionally, the comparison of the antenna produced in this article with the previously studied antennas is given in Table 5.

Graphene is preferred in antennas due to its many important potentials, such as reducing antenna sizes, providing good integration with electronic circuits, dynamic adjustment possibility, transparency, and flexibility [106]. The most important feature of graphene that makes it usable in antenna applications is the conductivity tensor in its structure [107,108]. The conductivity tensor depends on many parameters such as temperature, scattering rate, Fermi energy, and electron velocity. The conductivity tensor can be controlled when interacting with static electricity, and magnetic fields from the outside provide a dynamic structure for products made of graphene. Another important feature is that it is frequency-dependent. As the applied electric field increases, the conductivity values increase, but when the frequency rises above 103 GHz in the same electric field, it quickly decreases to zero [109]. Therefore, their conductivity changes according to frequency ranges. Since the frequency range used in this study is between 2 and 20 GHz, graphene is a good conductor in this range. Although copper offers better properties for antenna production, the fact that graphene and PLA can be used with a 3D printer provides a very good advantage in terms of production.

Gómez-Díaz and Perruisseau-Carrier carried out the first graphene-based patch antenna in their study [110]. They produced the antenna patch part from high-rate graphene. The return loss and radiation efficiency of the antenna were investigated, and it was observed that the efficiency decreased due to losses in graphene. Another important inference is that regardless of the externally applied electric field, it does not significantly change the resonance frequency of the patch. Therefore, there is no need to go through reconfiguration while antenna efficiency can be controlled.

In another study, an antenna with a diaphragm connection was produced to be used in the band from 14 to 16 GHz [109]. In this study, the patch is made of copper, and the surface resistance is 100 Ω. This property corresponds to the case where τ = 0.5 ps and $\mu_{c}$ = 0.175 eV in graphene. In the simulation study where graphene is applied, the return loss is quite good, while the radiation efficiency and radiation power are lower than the metal patch and no-patch conditions. To summarize, it is not very useful to use graphene to reduce the size of the antennas or change their geometry, but it is seen that it is mechanically flexible, has optical transparency, is easy to produce with a 3D printer, and even creating a conductive patch with graphene and metal mixture provides better performance. It is known that these examples are valid for simple monoatomic graphene layers, but this situation changes in multi-layer graphene applications with an increase in radiation efficiency [109].

## 4. Conclusions

In summary, three different bow-tie microstrip antennas that can operate in the 2–20 GHz range for ultra-wideband applications have been designed and experimentally implemented. Numerical analyses were made using the finite integration technique in the CST Microwave Studio simulation program. The designed antennas were selected from the non-slot state, and the radiating part was selected with two different slot sizes and produced from graphene filament. The substrate is produced from PLA material. It has been observed that all three designs perform quite well in ultra-wideband applications. The effect of design differences in antennas is reflected in antenna return losses. Especially for Antenna-1, the return loss seems to be quite good. In addition, copper is generally used for the radiating part in classical methods. This study has also shown that graphene performs better than copper because it is a good conductor, contrary to classical methods.

Another critical point is that the production of 3D-printer technology is one of the new aspects of the proposed study, as it is cost-effective and easy to produce, unlike classical production methods. In addition, the voltages applied to graphene changes the chemical potential and cause changes in the permittivity of graphene. In this respect, it has been a critical study to provide the opportunity to create different frequency ranges. It is considered to use one or more of these produced antennas for biomedical UWB applications for future studies.

## Figures and Tables

**Figure 1 polymers-13-03724-f001:**
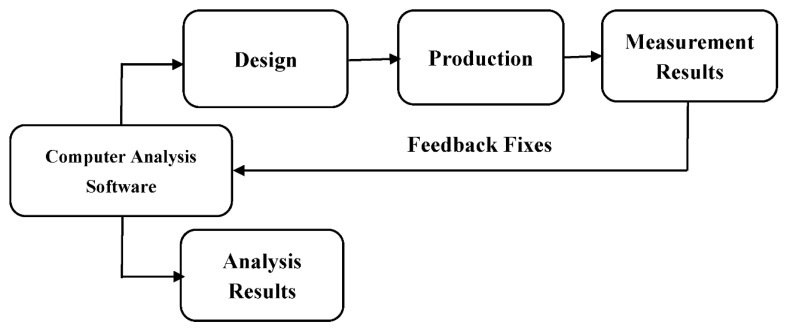
The design procedure of bow-tie microstrip patch antenna.

**Figure 2 polymers-13-03724-f002:**
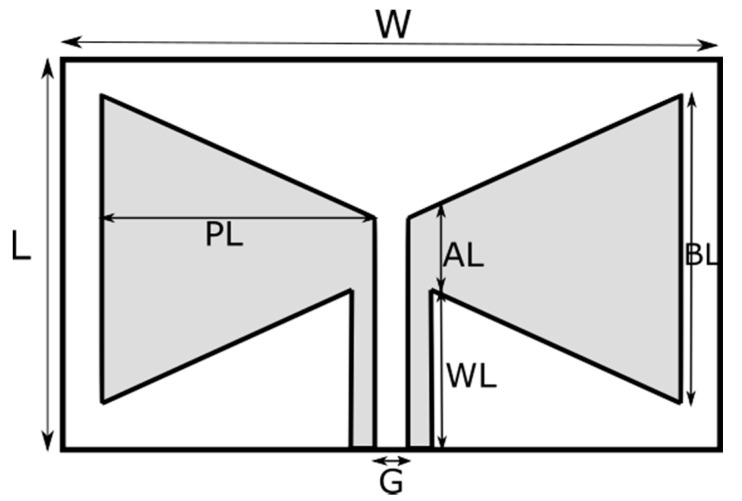
The design geometry of Antenna-1.

**Figure 3 polymers-13-03724-f003:**
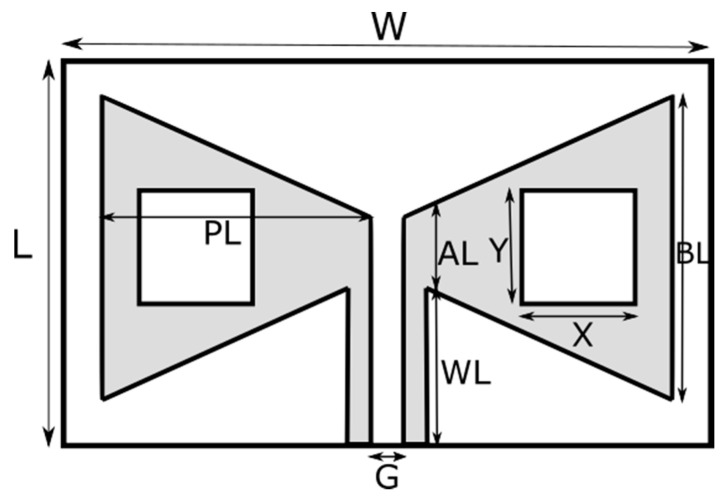
The design geometry of Antenna-2 and Antenna-3 with graphene substrate and different patch dimensions.

**Figure 4 polymers-13-03724-f004:**
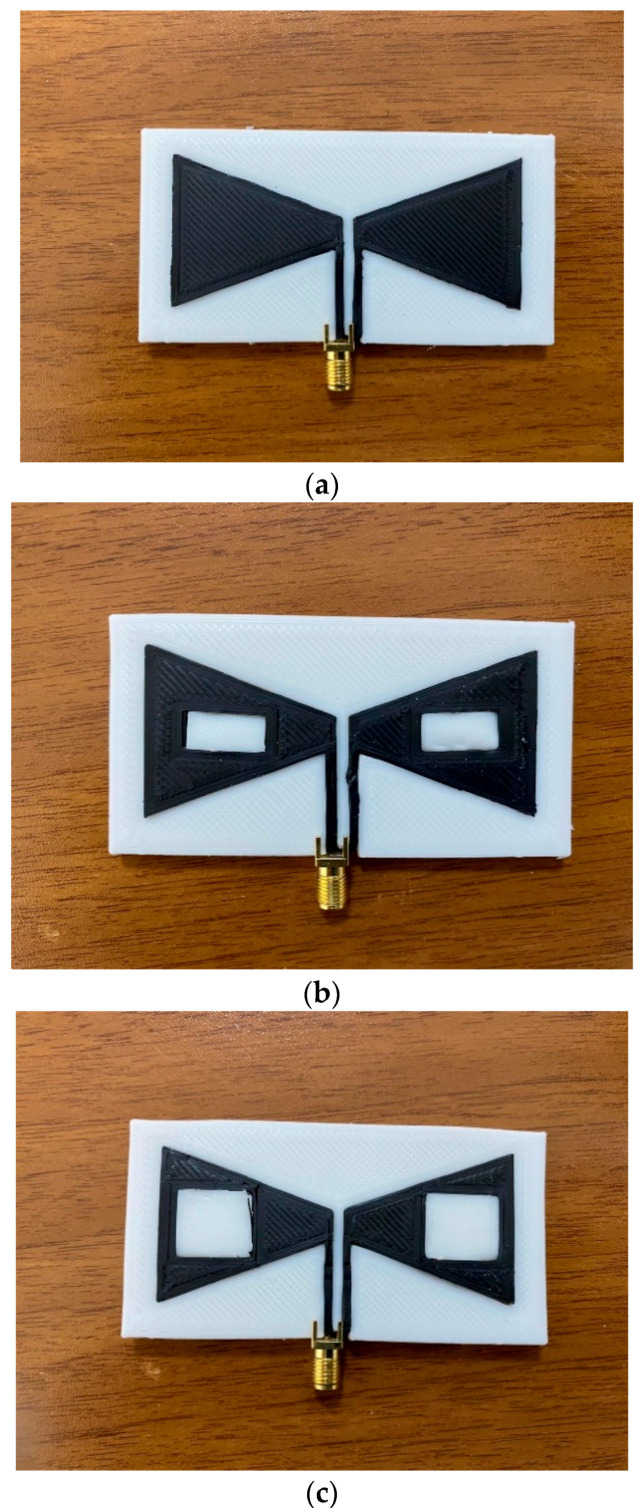
Fabricated bow-tie (**a**) Antenna-1, (**b**) Antenna-2, and (**c**) Antenna-3.

**Figure 5 polymers-13-03724-f005:**
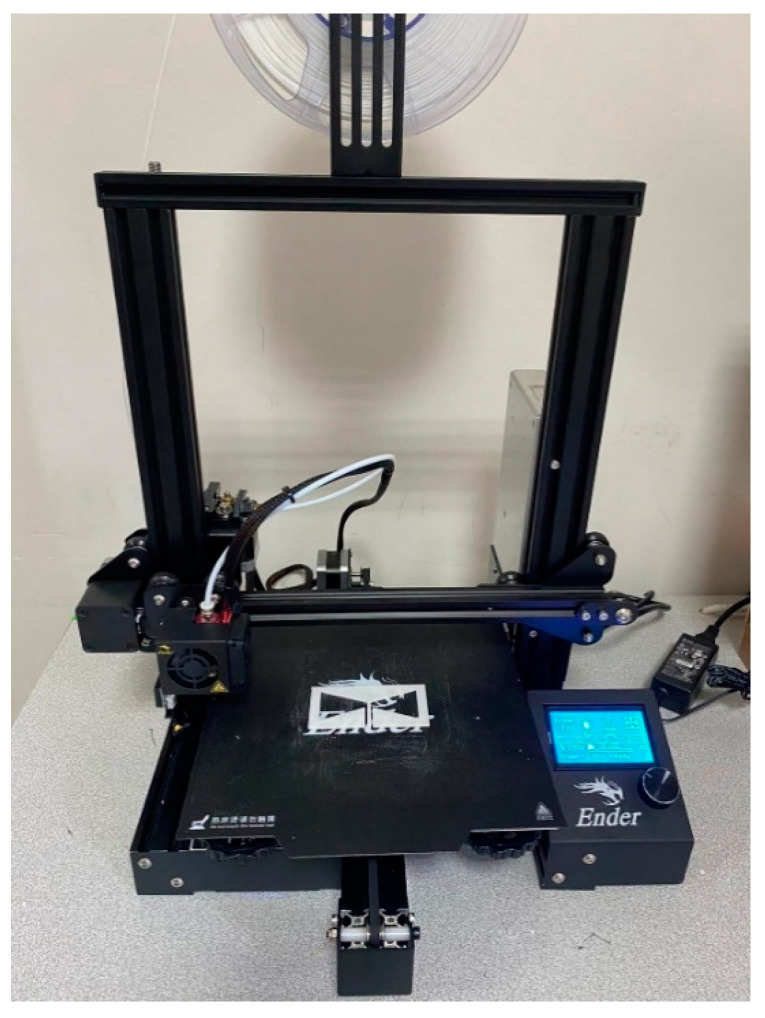
Creality Ender 3 Pro 3D printer.

**Figure 6 polymers-13-03724-f006:**
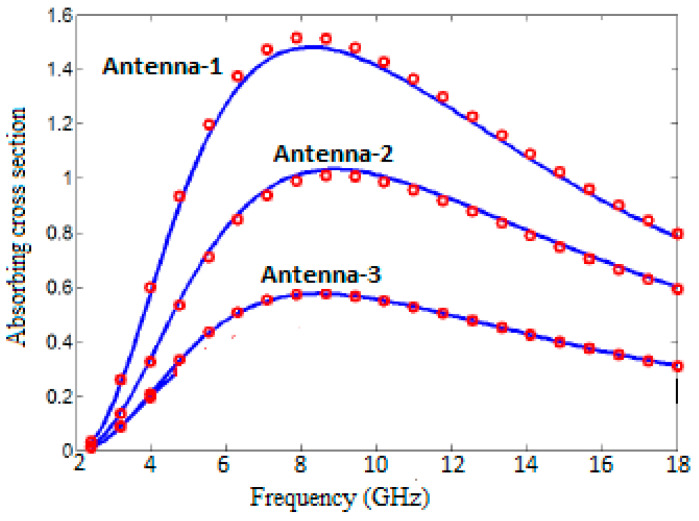
Absorbing cross-section in (µm)^2^ of Antenna-1, Antenna-2, and Antenna-3 with T = 300 K, $\mu_{e}$ = 0 eV in CST Microwave Studio.

**Figure 7 polymers-13-03724-f007:**
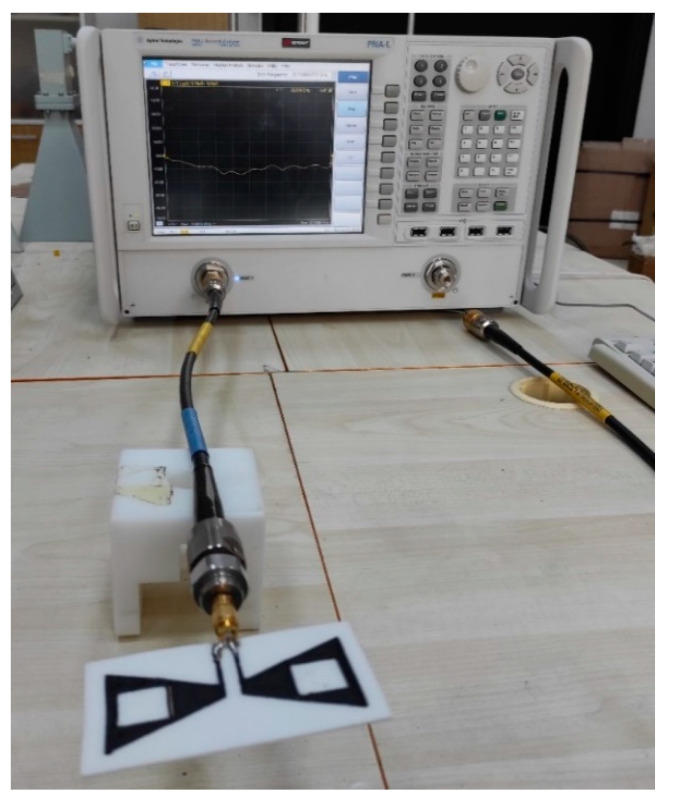
Measurement setup by using Keysight Technologies’ PNA-L Agilent vector network analyzer.

**Figure 8 polymers-13-03724-f008:**
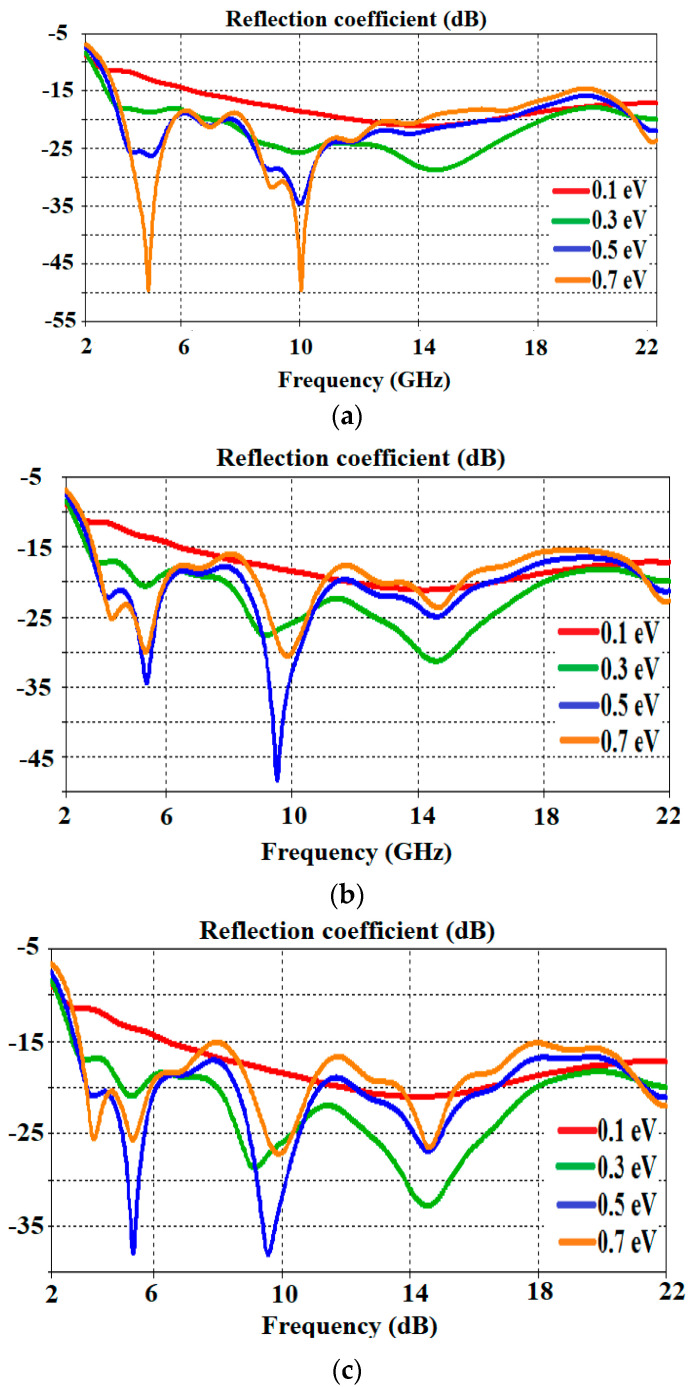
Reflection coefficient of (**a**) Antenna-1, (**b**) Antenna-2, and (**c**) Antenna-3 for varying chemical potential.

**Figure 9 polymers-13-03724-f009:**
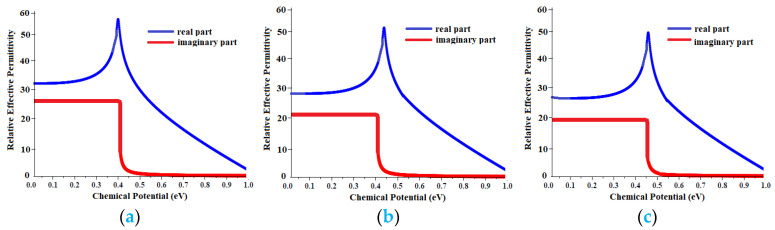
The real and imaginary parts of the graphene dielectric permittivity for (**a**) Antenna-1, (**b**) Antenna-2, and (**c**) Antenna-3 as the functions of the chemical potential.

**Figure 10 polymers-13-03724-f010:**
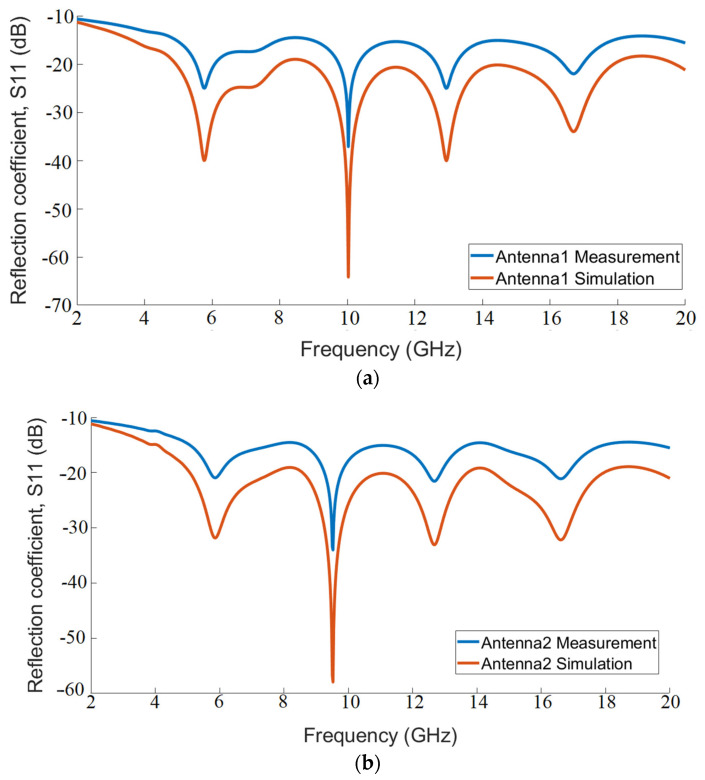
Reflection coefficients of (**a**) Antenna-1, (**b**) Antenna-2, and (**c**) Antenna-3 for bandwidth performance.

**Figure 11 polymers-13-03724-f011:**
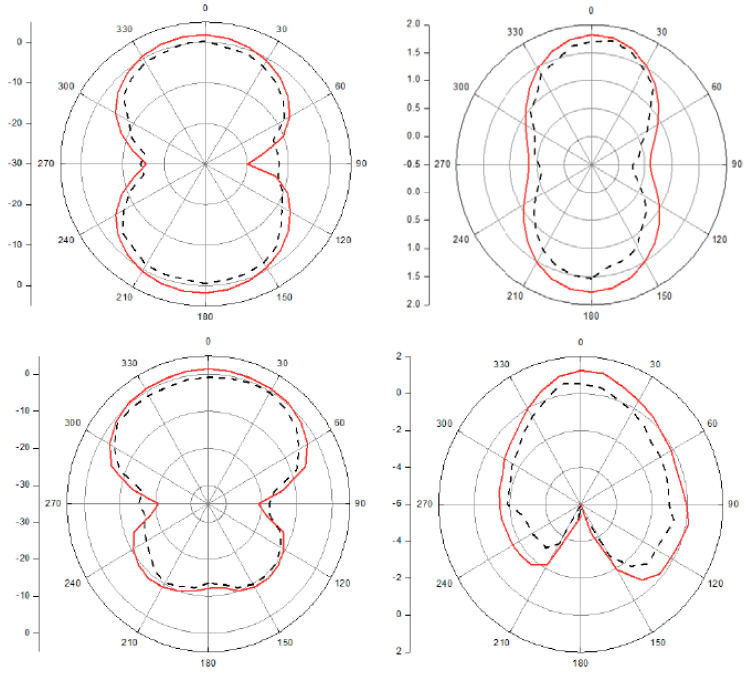
Radiation pattern measured (dotted-line) and simulated (straight-line) at frequency 9.5 GHz (E-plane graphics on the left side and H-plane graphics on the right side).

**Figure 12 polymers-13-03724-f012:**
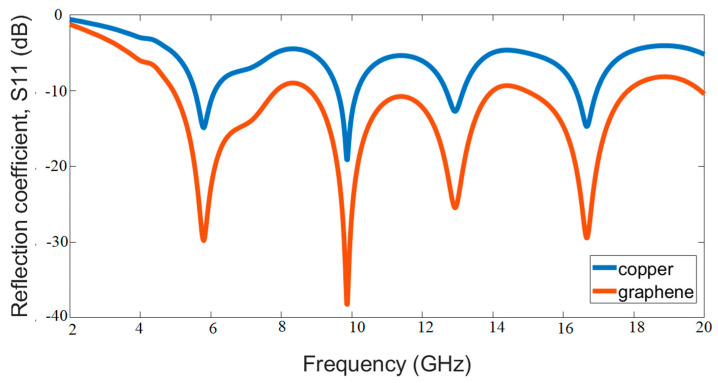
The simulation comparison results of the return loss (S_11_) of the graphene-based antenna and copper-based antenna for Antenna-1 in this study.

**Table 1 polymers-13-03724-t001:** Physical and print performance properties of Esun PLA+ filament [87].

Physical Performance Properties		Print Performance Properties	
Parameters	PLA+	Parameters	PLA+
Appearance	white	Surface finish	No obvious layering
Tensile strength	63 Mpa	Hole column fitting model	0.3 mm
Elongation at break	20%	Hanging model	<60°
Bending strength	74 Mpa	Suspension bridge	50 mm
Flexural modulus	1934 Mpa	-	-
Notch impact	9	-	-

**Table 2 polymers-13-03724-t002:** Properties of 3D-conductive graphene composite filament [88].

Properties	
Brand	Black Magic 3D
Weight	0.1
Volume resistivity	0.6 ohm-cm
Color	Black
Diameter	1.75 mm
Size	100 g
Min/max printing temperature	220 °C
Country	United States

**Table 3 polymers-13-03724-t003:** Bow-tie Antenna-2 and Antenna-3 parameters.

Parameters	Dimensions Antenna-2	Dimensions Antenna-3
Patch length PL	36 mm	36 mm
Base length BL	32 mm	32 mm
Gap	4 mm	4 mm
Feedline width	0.6 mm	0.6 mm
Apex length AL	5.4 mm	5.4 mm
Substrate length L	44 mm	44 mm
Substrate width W	88 mm	88 mm
Aperture length for patch X	8 mm	14 mm
Aperture width for patch Y	16 mm	16 mm

**Table 4 polymers-13-03724-t004:** Performance comparison of previous studies and graphene-based bow-tie antennas designed in this study.

Performance Parameters	Designed Bow-Tie Antennas			Conventional Antennas	Graphene-Based Antennas		
	Ant.1	Ant.2	Ant.3	[98]	[99]	[78]	[100]
Return Loss (dB)	−38.85	−34.28	−29.95	−32.50	−64.00	−25.23	−39.92
VSWR	1.06	1.13	1.00	1.06	1.04	NR	NR
Peak Gain (dBi)	9.10	8.70	8.90	5.4	7.55	4.83	NR
BandWidth (GHz)	18	18	18	8.2	8.45	7.35	4.0

**Table 5 polymers-13-03724-t005:** Comparison of the manufactured antenna with some studies.

Reference No.	Antenna Type	Dimensions mm^3^ and Applications	Substrate, Conductive Element	Bandwidth (FBW%)	Resonant Frequency	Antenna Gain and Efficiency
[102]	Elliptical quasi-dipole antenna	46 × 45 mm^2,^ 2 up to 5 GHz for low-cost wireless communications applications.	Kapton Polyimide, graphene flakes	1–5 GHz	2 GHz	2.3 dBi at 4.8 GHz, 56 ± 5%
[103]	CPW-fed H-shaped slot antenna	32 × 52 × 0.28, UWB antenna for wearable applications	Flexible ceramic substrate, graphene assembled film (GAF)	4.1–8.0 GHz	4.45, 5.6, and 7.1 GHz	3.9 dBi at 7.45 GHz
[104]	CPW-fed rectangular slot with chamfer	11.8 × 12.2 × 0.1, Mobile terminal for fifth generation (5G)	Kapton polyimide, Graphene ink	14.30–15.71 GHz	13.8 GHz	9.28 dBi, 67.44%
[105]	Bow-tie antenna	Optical near-field enhancement	Glass graphene	14.3–35.3 THz	-	-
This study	Bow-tie antenna	44 × 88 (outer size), ultra-wideband applications for especially medical applications	PLA, graphene filament	2–20 GHz	Antenna-1 10.01 GHz, Antenna-2 9.6 GHz, Antenna-3 9.8 GHz	Antenna-1 9.10 dBi, Antenna-2 8.70 dBi, Antenna-3 8.90 dBi

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
