# Peer review of "3D-Printed Graphene-Based Bow-Tie Microstrip Antenna Design and Analysis for Ultra-Wideband Applications"

_polymers, 2021, doi:10.3390/polym13213724_

Round 1

Reviewer 1 Report

The manuscript entitled “Ultra-wideband 3D Printed Graphene-Based Bow-Tie Microstrip Antenna Design and Analysis for Biomedical Applications” was submitted by Aydin in the Polymers. The author investigated the Ultra-wideband microstrip Bow-Tie antenna using graphene for biomedical applications. This study also shows how the use of graphene allows changing the antenna’s operating frequency dynamically and that the antennas can be produced easily with a standard 3D printer without compromising their performance. Overall, the manuscript is well-written with interesting results. However, it can be further improved. I have the following suggestions and comments that need to be addressed.

  1. The title focuses on biomedical applications, but the author generally mentioned that this antenna can be applied for medical applications. Unless this antenna has not been applied to some device in the proposed study, I believe the title does not justify the results.
  2. Is there any technical reason for specifically choosing this design? If we replace this design with square or circular patches, then what are the implications?
  3. Please mention the conflict between the simulated and measured results of the proposed antenna.
  4. Why did the author decide to compare the graphene-based antenna with other types of antennas? It would be better if there was a relevant comparison in Table 3. It needs clarification.
  5. The language of the manuscript requires thorough revision with typo-free text. Also, it is suggested to modify the symbol S11 and replace ‘ohm’ with the symbol. Similarly, please improve the captions of Figures 5 and 7. It should be technical with some information.
  6. In the conclusion section, a lot of information is redundant. From lines#382-385, it seems like an introduction. Please be specific and thoroughly revise the conclusion.
  7. Some important studies are discussing the methods to enhance the bandwidth of antennas and metamaterial-based absorbers. Please include in the introduction: “Modified Wang Shaped Ultra-Wideband (UWB) Fractal Patch Antenna for Millimetre-Wave Applications” “Elliptical metallic rings-shaped fractal metamaterial absorber in the visible regime”.

Author Response

Dear Reviewer,

I have revised the paper as your suggestions and uploaded as a revised submission.

Thank you very much for your valuable comments and time. Please see the attachment.

Best regards,

Dr. Emine AVŞAR AYDIN.

Author Response

(The authors gave the same response as above.)

Round 2

Reviewer 1 Report

Article can be accepted now for the publication. 

Reviewer 2 Report

Dear editor

The authors approximately answered my comments and now it can be published.